# One-Step Oxidation of Orange Peel Waste to Carbon Feedstock for Bacterial Production of Polyhydroxybutyrate

**DOI:** 10.3390/polym15030697

**Published:** 2023-01-30

**Authors:** Maryam Davaritouchaee, Imann Mosleh, Younas Dadmohammadi, Alireza Abbaspourrad

**Affiliations:** Department of Food Science, College of Agriculture & Life Sciences, Cornell University, Stocking Hall, Ithaca, New York, NY 14853, USA

**Keywords:** waste to value, orange waste, oxidation treatment, limonene degradation, sugar extraction, bacterial growth, polyhydroxybutyrate biopolymers

## Abstract

Orange peels are an abundant food waste stream that can be converted into useful products, such as polyhydroxyalkanoates (PHAs). Limonene, however, is a key barrier to building a successful biopolymer synthesis from orange peels as it inhibits microbial growth. We designed a one-pot oxidation system that releases the sugars from orange peels while eliminating limonene through superoxide (O_2_^• −^) generated from potassium superoxide (KO_2_). The optimum conditions were found to be treatment with 0.05 M KO_2_ for 1 h, where 55% of the sugars present in orange peels were released and recovered. The orange peel sugars were then used, directly, as a carbon source for polyhydroxybutyrate (PHB) production by engineered *Escherichia coli*. Cell growth was improved in the presence of the orange peel liquor with 3 *w*/*v*% exhibiting 90–100% cell viability. The bacterial production of PHB using orange peel liquor led to 1.7–3.0 g/L cell dry weight and 136–393 mg (8–13 *w*/*w*%) ultra-high molecular weight PHB content (*Mw* of ~1900 kDa) during a 24 to 96 h fermentation period. The comprehensive thermal characterization of the isolated PHBs revealed polymeric properties similar to PHBs resulting from pure glucose or fructose. Our one-pot oxidation process for liberating sugars and eliminating inhibitory compounds is an efficient and easy method to release sugars from orange peels and eliminate limonene, or residual limonene post limonene extraction, and shows great promise for extracting sugars from other complex biomass materials.

## 1. Introduction

Petroleum-based polymers and plastics are an environmental hazard due to their production, their poor recyclability, and the waste that lingers for decades in landfills. Additionally, at all levels the processes associated with these materials release greenhouse and toxic gases into the environment [1]. An effective process to replace petroleum-based polymers is using natural, plant-based, carbon-based feedstocks to make bio-based polymer syntheses such as polyhydroxyalkanoates (PHAs). Using carbon-based feedstocks, several different bacterial strains have been found to accumulate PHAs as energy-storage compounds in the form of intracellular granules [2,3]. Unfortunately, the carbon sources used for PHA production via bacterial fermentation is cost-prohibitive, comprising almost 50% of the overall cost [4]. To increase sustainability within the biopolymer industry, upstream costs must be reduced and one way to do this is to leverage food and agricultural waste as sources of starting materials [5,6,7,8,9]. The majority of food and agriculture wastes contain cellulose, hemicellulose, lignin, proteins, lipids, and a small amount of ash (inorganic oxides) [10,11]. Cellulosic sugars can be used to produce biopolymers and other waste components directly or indirectly and may lead to better yields or novel properties. Cellulosic materials are recalcitrant, however, and pretreatment is required before using them as feedstocks. 

A major barrier to the growth of the biobased industry based on agricultural and food waste is the lack of effective pretreatment processes that can transform cellulosic materials into useable sugar molecules. Organisms, such as termites and fungi, have been shown to convert carbon feedstocks into sugar molecules via oxidation pathways [12]. The enzymes involved in the oxidation reaction of complex materials are lignin peroxidases, manganese peroxidases, aldo-keto reductases, alcohol dehydrogenases, glucose dehydrogenases, superoxide dismutase, and catalase. These enzymes undergo reactions that involve radical species such as superoxide anion radicals and hydroxyl radicals [13,14]. Inspired by these species, we developed a single pot chemical oxidation method to treat orange peel waste and produce viable carbon-feedstocks for PHB production. 

Oxidation processes are commonly used as pretreatment methods to enhance the bio-conversion productivity of waste stream for biofuel production [15]. Specifically, radical oxidation methods are mild, selective, green, and less energy intensive compared with other pretreatment techniques. Superoxide radical anion, in particular, has been shown to break linkages in complex structures such as cellulose [16,17]. Here, for the first time, we report the use of an oxidation method to deconstruct the carbohydrate structures of orange peels to produce sugars suitable for PHB production.

Orange peel waste is a promising nutrient source for PHB producing bacteria [18]; it contains soluble and insoluble carbohydrates and a low protein content leading to a high carbon to nitrogen ratio [19,20]. With an annual global production of 10 million tons, orange peel waste is the predominant waste from citrus processing industries [16,17,18]. While scarce, there are some reports using orange peel waste to produce PHB [21,22]. The methods described focus on water soluble sugar monomers in orange peel to be converted to biopolymer with bacteria, carbohydrates of orange peel as the nutrient-rich broth for bacteria, and orange peel to increase bacteria culture media C/N ratio. To date, orange peel conversion to PHB has been investigated to improve fermentation conditions (pH, temperature, and fermenter setup) but not how to extract the sugars and limit inhibitory compounds most efficiently. Complicated or low efficiency pretreatment steps requiring isolation and purification need to be avoided if wide scale adoption is to be realized.

Although a few different physicochemical approaches have been used to convert orange peel waste into value-added products [19,22,23], the complexity and the toxicity of some of the compounds in orange peels have prevented these products and processes from becoming widely available [24,25]. For example, residual limonene, a cyclic monoterpene and the main constituent of orange essential oils, possesses an inhibitory effect on microbial growth [20,26]. Residual limonene concentrations as low as 1 *v*/*v*% can decrease the growth rate among different bacteria strains by 60%. To be able to use the sugars from orange peels as a carbon-food source for bacteria, residual limonene must be removed. Ideally, this would be a single pretreatment step that would be mild, low energy, and with no impact on the yield of the sugars [27]. 

We report here a one-pot oxidation treatment process where orange peel waste is broken down into carbon feedstocks suitable as bacterial nutrient sources, while simultaneously degrading any limonene present. Once the orange peels have been oxidized, the spent peels are collected by filtration and the liquor is freeze dried and then used, as is, to promote the growth and production of PHB using bacterial fermentation. Bacteria that produce PHB include recombinant *Escherichia coli JM109 (E. Coli),* and both gram-positive and gram-negative bacteria. Some examples are *Cupriavidus necator*, *Alcaligenes* spp., *Bacillus* spp., *Nocardia* spp., and *Pseudomonas* spp., and *Rhizobium* spp. *E. Coli* was chosen as the bacteria of choice because it is safe and a good host for gene cloning. 

## 2. Materials and Methods

### 2.1. Chemicals and Materials

Orange peel waste was donated by Tropicana PepsiCo (Valencia from Florida Hamlin variety samples received directly from the extractor). Cellulose, mannose, xylose, arabinose, galactose, sucrose, glucose, pectin from citrus peel, galacturonic acid, isopropyl-β-D-thiogalactoside (IPTG), furfural, hydroxymethylfurfural (HMF), potassium superoxide, ampicillin sodium salt, and Luria–Bertani (LB) Broth (Lennox) were purchased from Sigma-Aldrich. Fructose was obtained from Spectrum Chemical. In all experiments, ultra-pure milli-Q water (MQ) was purified to 18 MΩ·cm. Solvents used included tetrahydrofuran (THF), methanol, acetone, and chloroform which were purchased from Sigma-Aldrich. All reagents were reagent grade and used without any further purification unless otherwise specified.

### 2.2. Oxidative Pretreatment of Orange Peels by Superoxide (O_2_^• −^) 

Orange peel waste was dried at 65 °C for 48 h, then chopped, ground, and passed through a Tyler standard screen scale, No. 20 sieve, resulting in fine particles with a size range of 0.3–0.8 mm. 

The oxidation of orange peel particles was conducted in the presence of potassium superoxide reagent to generate O_2_^• −^ species as reported in a previous study [12]. Dried and powdered orange peel particles (10 g) were oxidized with different amounts of potassium superoxide (1, 0.50, 0.25, 0.05, and 0.02 M) in 500 mL of an alkaline aqueous media (pH ~10) at room temperature. The oxidation process was conducted for 1, 2, and 3 h, under continuous stirring at 400 rpm. The treated samples were washed repeatedly with MQ water and the brown liquid extracts were collected by filtration and neutralized by either concentrated NaOH and H_2_SO_4_ as needed. An aliquot was saved for sugar analysis, and the remaining sugar solution was freeze-dried and used as the feedstock in the production of PHB. 

The quantity and type of released sugars and inhibitory compounds contained in the liquor after orange peel oxidation was quantified by an HPLC (Agilent Technologies 1100) equipped with an Aminex HPX-87H column (Bio-Rad, Hercules, CA, USA). Degassed sulfuric acid (0.01 N) was used as the mobile phase at a flow rate of 0.4 mL/min at 40 °C. A refractive index (RI) detector was used for sugar analysis based on the area under the peak and the retention time of the standards (reagent grade sugars). 

To characterize the removal of limonene after oxidation, the tip of a glass rod was dipped into the superoxide radical anion solution and the mass of limonene was monitored by DART-MS. Two spectra were taken before and after the oxidation reaction in real-time by Thermo Scientific Exactive DART-Orbitrap under nitrogen atmosphere at 200 °C.

### 2.3. General Procedure to Produce PHB from Controls and Oxidized Orange Peel Waste 

PHB production was evaluated by engineered *E. coli*, *JM109 (source Promega Corporation)*, in a shaking flask fermenter containing four different carbon sources: glucose, fructose, simulated orange peel sugar for comparison, and orange peel liquor. Fermentations were carried out for 24, 48, 72, and 96 h. To understand the effect of oxidation on the orange peels with respect to fermentation performance, bacterial growth, and PHB production, we designed three different control experiments: (1) glucose, (2) fructose, and (3) simulated orange peel sugar. 

Glucose and fructose were chosen as the primary sugars present in orange peels. The content of orange peels, in dry weight, has been reported in the literature as containing 15–25 *w*/*w*% pectin, 20–40 *w*/*w*% soluble sugars, 20–35 *w*/*w*% carbohydrate complexes, 5–10 *w*/*w*% protein, 3–5 *w*/*w*% starch, 1–7 *w*/*w*% lignin, as well as 2–5 *w*/*w*% organic acids and essential oils [28,29,30]. Glucose and fructose were chosen as controls because they are the primary soluble sugars present in orange peels. The simulated orange peel sugar was made to probe any synergy between the carbohydrate-based components of the orange peels and was prepared to mimic the literature reports by combining 20 *w*/*w*% pectin, 25 *w*/*w*% cellulose, 25 *w*/*w*% glucose, 20 *w*/*w*% fructose, and 10 *w*/*w*% sucrose, arabinose, galactose, xylose, mannose, and glucuronic acid. Our experimental carbon source sample, orange peel liquor, was prepared by oxidizing orange peels using a potassium superoxide (KO_2_) reagent.

After each fermentation period, PHB were extracted from the cells. Subsequently, the obtained polymer was characterized by NMR, gel permeation chromatography (GPC), and differential scanning calorimetry (DSC). The overall schematic of the experimental work is illustrated in Figure 1.

### 2.4. Bacterial Strains, Plasmid Transformation, and Growth Kinetics

A plasmid harboring phaCAB DNA sequence, amplified from the *R. eutropha* H16 genomic DNA resistant to ampicillin was purchased from Biomatik Corporation (Cambridge, Canada) which was designed according to *Escherichia coli* codon preferences. The DNA fragment of the phaCAB operon consisting of phaA (3-ketothiolase), phaB (acetoacetyl-CoA reductase), and phaC (PHA synthase) genes, with ribosomal binding sites, in addition to an ampicillin resistance marker and T7 promoter, were inserted into a pUC57 plasmid cloning vector. The pUC57 plasmid cloning vector was isolated from *E. coli* strain DH5α. The constructed plasmid was transferred to *E. coli* JM109 competent cell with heat shock according to the manufacturer’s protocol (Promega Corporation). The resulting cells with different dilution ratios were transferred to an agar plate containing ampicillin and were incubated overnight at 37 °C [31].

A colony of *E. coli* JM109 cells were inoculated in LB medium to ensure cell growth. Additionally, the medium was supplemented with 100 mg/L ampicillin. To study the cell viability in the presence of different sugar sources, first, the bacterial strain was cultured for 18 h at 37 °C with 250 rpm reciprocal shaking. Then 0.3 mL of the culture was transferred to 25 mL flasks supplemented with 10 mL of fresh LB and 0.5, 1, 1.5, 2, 3, and 5 *w*/*v*% of each considering carbon sources. The OD was measured using the live cells to total cells ratio at 550 nm (OD_550_) every 30 min for 7 h and the absorbances were compared with the control (0 *w*/*v*% carbon source). The studied strain reached maximum growth rate in 7 h.

To produce PHB, 2 mL of cultured cells were transferred into 1 L of LB containing 100 mg/L ampicillin. IPTG, mimics allolactose, with a concentration of 100 mg/L was added to the culture media to induce gene expression in the lac operon [32,33] when the OD_550_ reached 0.6–0.8 (roughly 9 h). After one hour, the culture media was supplemented with 2 *w*/*v*% of the corresponding carbon source, and the fermentations were carried out for 24, 48, 72, and 96 h at 37 °C and 250 rpm. Samples were taken out periodically to determine the OD_550_. The optical density of culture media was measured using the SpectraMax iD3 plate reader. All fermentation experiments were carried out in triplicate.

At the end of each fermentation period, cells were collected by centrifugation at 4500× *g* for 30 min at 4 °C. The cell pellets were washed twice with MQ water to remove any remaining nutrition media. The resulting cells were then lyophilized and weighed. The intracellular PHB from each set of samples was extracted according to the published process with slight modification [34] using the chloroform dispersion method in which dried cells are immersed in chloroform. After vigorous shaking for one day at room temperature, cell debris was filtered out and the PHB-containing supernatant was concentrated and the PHB was precipitated using cold methanol.

### 2.5. Characterization Methods

Structural analysis of the extracted PHB was carried out using Nuclear Magnetic Resonance (NMR) Analysis. The spectra were recorded using Bruker 500 MHz using deuterated chloroform (CDCl_3_) as the solvent. Solid-state ^13^C spectra of raw materials and treated orange peels were acquired on a 500 MHz Varian INOVA spectrometer equipped with a 3.2 mm Phoenix HX probe head (Phoenix NMR, Loveland, CO) using standard-wall zirconia rotors spinning at 20 kHz. Spectra were collected with CP-MAS using a linear ramp on ^1^H as provided in the Varian “tancpx” pulse sequence, and TPPI ^1^H decoupling, and 1500–2000 scans were averaged using 3 ms CP contact time, 20 ms acquisition time, and 4 s relaxation delay.

FTIR with Attenuated Total Reflection (ATR) module was used to study functional groups and structural changes in the oxidized orange peel. FTIR spectra were obtained on a Shimadzu IR-Affinity-1S FTIR spectrophotometer equipped with an ATR module with 64 scans in absorbance mode. Samples were analyzed from 600 to 4000 cm^−1^ at a resolution of 8 cm^−1^.

The weight-average molecular weight (*Mw*), number-average molecular weight (*Mn*), molecular weight in the peak maximum (*Mp*) of chromatogram curve, and polydispersity index (*PDI*, *Mw*/*Mn*) of generated PHB were determined by a chloroform-based Gel Permeation Chromatography (GPC) with 0.25% Triethylamine (TEA) equipped with two Agilent, PolyPore (300 × 7.5 mm) columns. Samples (10 mg/mL in chloroform) were analyzed using Waters Alliance HPLC System Pump with Waters 2410 Differential Refractometer (RI) at 35 °C and with a flow rate of 1.0 mL/min. The column system was calibrated with a set of monodisperse polystyrene standards.

To analyze the thermal properties of the PHB polymer such as melting temperature (T_m_), glass transition temperature (T_g_), melting enthalpy (ΔH_m_), and crystallinity (X_c_) of the PHB samples obtained from different sugar sources and with various incubation times. Differential Scanning Calorimetry (DSC) was performed using Q2500 TA Instruments. Approximately 5 mg of PHB from each fermenter was heated in an aluminum weighing crucible with heating and cooling rates of 10 °C/min to 200 °C in a nitrogen environment. The first heating, cooling, and the second heating cycles were operated from room temperature to 200 °C, from 200 °C to −50 °C, and from −50 °C to 200 °C, correspondingly. The degree of crystallinity was calculated by dividing the area under the melting peak of prepared PHB by melting enthalpy of 100% crystalline PHB.

Bacterial cells were prepared for TEM analysis by harvesting after the specified incubation times, pelletizing, and washing three times with phosphate-buffered saline (PBS), and then fixed with 2.5% (*v*/*v*) glutaraldehyde in PBS overnight at −8 °C. The cells were dehydrated by serial dilution in ethanol. A FEI F20 TEM STEM (Transmission Electron Microscopy), 200 kV field emission transmission electron instrument with a monochromator, equipped with a Gatan tridium spectrometer at high energy resolution < 0.2 eV was used to capture the TEM images of the PHB granules formed within the cell. 

### 2.6. Statistical Analysis

JMP software (Pro 16) was used for statistical analysis by two-way ANOVA post-hoc Tukey’s HSD test. 

## 3. Results and Discussion

### 3.1. Chemical Composition of Orange Peels after Oxidation

In this work, different oxidation conditions were studied to develop a pretreatment protocol for orange peels to improve sugar release and degrade limonene. The composition of the orange peel liquor after oxidation was quantified using HPLC analysis (Figure 2 and Appendix A). Intact orange peels, that is, peels from which the essential oils were not yet extracted, were used. Ideally, these peels would have the essential oils extracted downstream or upstream from this step. Either way, the oil extraction process would not have an impact on the sugars, and by using intact peels we have demonstrated that even at high starting levels of limonene our oxidation successfully eliminates limonene. 

In the control sample, orange peels treated with dilute alkaline solution (0.05 M NaOH) generated 6.26 g/L glucose, 1.72 g/L fructose, 0.01 g/L acetic acid, and 9.17 g/L galacturonic acid. Galactose/xylose, hydroxymethylfurfural (HMF), and furfural were not detected in the control sample at any time. In oxidative runs, the optimum conditions for extracting the maximum sugar quantity were achieved using 0.05 M KO_2_ (0.05-OX-OP) after 1 h. The major sugars released from the optimum run after 1 h were glucose (11.19 g/L), galactose (0.62 g/L), xylose (0.54 g/L), fructose (4.97 g/L), and galacturonic acid (4.46 g/L). An optimum oxidation treatment also produced a negligible number of inhibitory compounds such as acetic acids, furfural, and HMF. The yield of sugars released after 1 h of each treatment was 40% (control (alkaline solutions no oxidant)), 75% (0.02-OX-OP), 87% (0.05-OX-OP)*,* 54% (0.25-OX-OP), 29% (0.5-OX-OP), and 32% (1-OX-OP). In addition, the yields of fermentable sugar monomer and inhibitory compounds after 1, 2, and 3 h of oxidation are presented in Appendix A.

Increasing pretreatment time did not improve sugar release (Appendix A). For example, increasing the time from 1 h to 3 h for sample 0.05-OX-OP caused free sugars to degrade and increased the concentration of inhibitory compounds. Upon longer oxidation from 1 h to 3 h, the glucose and fructose availability for 0.05-OX-OP decreased by 43% and 56%, respectively. Additionally raising the concentration of KO_2_ from 0.05 M to 0.25 M led to a 38% drop in sugar quantity after 1 h. Any further increase of the KO_2_ concentration, such as to 0.5 M or 1 M (0.5-OX-OP and 1-OX-OP), produced much lower yields of sugar content.

Further, the oxidation method we used was compared with acid hydrolysis, the most commonly used method for the polysaccharide degradation of orange peels [35,36]. Acid hydrolysis was carried out with 10 g of orange peel powder in an autoclave with 6 *w*/*v*% sulfuric acid at 120 °C for 1 h. Acid hydrolysis generated 6.95 g/L glucose, 0.60 g/L galactose, 0.35 g/L xylose, 7.87 g/L fructose, 0.27 g/L acetic acid, 1.11 g/L HMF, and 0.04 g/L furfural (Appendix A). The yield of sugar release was 78%, 10% lower than our optimized oxidation method. Further acid hydrolysis decreased glucose production by 37%, and led to 58% more fructose and 250% more inhibitory compounds (acetic acid, HMF, and furfural), compared with 0.05-OX-OP after 1 h of treatment. A full description of this data can be found in Appendix A. 

The main advantages of our oxidation pretreatment are a high released sugar content, a low inhibitory compound formation, a low energy input, a short treatment time, and green reaction conditions. 

### 3.2. Structural Alteration Study of Orange Peels with FTIR

FTIR analysis was used to qualitatively determine changes in the functional groups in the solid residue of oxidized orange peel compared with untreated orange peel (Figure 3). All samples were dried at 50 °C for 24 h to minimize the side effect of moisture on FTIR spectra.

The characteristic lignin bands, and bands corresponding to small organic compounds (methyl or methylene bands), increased in intensity after oxidation. The bands indicating the O-H groups present in the carbohydrates and lignin appeared at 3300 cm^−1^ in both control and treated samples. Any increase in the hydroxyl group stretching band was attributed to moisture content and the formation of phenolic derivatives [12]. The O-H groups are also present in cellulose and hemicellulose moieties, thus a decrease in the hydroxyl group band in the solid residue indicated the successful release of polysaccharides into the liquor. An increase in the intensity of the band at 2900–3000 cm^−1^ corresponded to CH_3_ and CH_2_ stretching vibrations which indicate free C-H groups in the sample. Lignin bands at 1300 and 1600 cm^−1^ were observed in the control sample, but their intensity was increased in the oxidized sample. This indicates the removal of polysaccharides and that lignin remained in the solid residue. A cellulose and hemicellulose band at 1030 cm^−1^ corresponding to C-O-C and C-O-H vibrations implied the successful release of sugar from the orange peels. The band around 1726–1733 cm^−1^ (carbonyl C=O stretching) weakened in the oxidized sample, signifying the removal of pectin ester linkages and the acetyl group of hemicellulose and their successful relocation to the sugar-rich media [37].

### 3.3. C Cross-Polarization Magic Angle Spinning (^13^C CP/MAS)

Normalized ^13^C CP/MAS NMR analysis with identical amounts of the sample was used to study the orange peels structural alteration using the 0.05-OX-OP oxidation treatment. Dried and ground orange peel and water-treated orange peel were prepared as control samples. The water-treated orange peel sample was prepared to remove the soluble sugars from the peels. These controls and the oxidized orange peel residue were analyzed by ^13^C solid-state NMR (Figure 4). 

Peaks associated with cellulose, hemicellulose, pectin, lignin, and essential oils appeared in the spectra of the control samples similar to those reported by Merino et al. [38] and Foston et al. [39]. Changes in peak intensity were modest when spectra of dried powder orange peels were compared with water-treated orange peels. The major difference was the sugar peak intensities which were present, but to a lesser extent for water-treated orange peels. We attribute this reduction to the solubilization of free sugar and its removal during the filtration process. The spectra of both control samples demonstrated the existence of citral, a monoterpene in essential oils, at 200.34, 165.16, and 132.18 ppm [40]. The peaks attributed to citral were reduced in the oxidized orange peel indicating a successful degradation or modification of this component. Additionally, the limonene peaks appeared in dried, water-treated samples (controls), as well as oxidized treated orange peels at 145.85 and 145.09 ppm. The removal of residual limonene from the orange peel liquor after treatment was further confirmed by the DART-MS study (Figure 5).

The decrease in cellulose and hemicellulose related peaks for the oxidized orange peel samples at 106.20 and 101.10 ppm (C_1_ in glucose unit), 89.70 and 85.20 ppm (C_4_ in glucose unit), 73.52 ppm (C_2,3,5_ in glucose unit), and 66.02 ppm (C_6_ in glucose unit) indicated the release of polysaccharide after oxidation. Additionally, ^13^C solid-state NMR can be used to determine the ratio of crystalline to amorphous cellulose which are represented by peaks at 89.70 and 85.20 ppm, respectively, in the control samples. The disappearance of C_4_ of crystalline cellulose in the oxidized orange peel sample provided evidence of polysaccharide degradation and the effectiveness of oxidation treatment.

Peaks for the carbonyl carbons associated with galacturonic acid and the acetyl groups of hemicellulose appeared in all controls in the range of 170–177 ppm. The intensity of these peaks decreased in the oxidized samples. The methoxy group of pectin was at 54.90 ppm decreased significantly with oxidation, more evidence of its successful degradation as observed by FTIR. The aliphatic peaks in a range of 19 to 32 ppm correspond to free alkyl groups and the 100 to 160 ppm range are specific to lignin signals which are frequently broad with low intensity. The intensity of correlated peaks for lignin is almost negligible, as expected, due to the low content of lignin in orange peels.

### 3.4. Bacterial Growth Kinetics with Oxidized Orange Peel Liquor and Limonene 

The bacterial growth was measured to study any inhibitory effects on cell growth posed by treated orange peel liquor by acquiring the optical density (OD) of the cells which is proportional to the number of cells per unit volume. The cell growth was compared with the control in which the *E. coli* strain was cultured in LB media with no exogenously added carbon sources.

The effect of limonene on the cell growth of JM109 was studied by incubating cells in LB with different concentrations of limonene added exogenously. At limonene concentrations above 0.5 *v*/*v*% a strong inhibitory effect on the kinetics of cell growth after 1.5 h was observed (Appendix A). Our results are in agreement with previous reports on the negative effect of limonene on the growth of *Cupriavidus necator H16* (and other strains), specifically, 1 *v*/*v*% limonene resulted in a 25% growth reduction [25,41,42]. Limonene interferes with the functioning of microorganisms by breaking the osmotic equilibrium of microorganisms at the cytoplasmic membrane [43,44]. Based on this, to produce PHB, the residual limonene component of the orange peel needed to be removed before fermentation. 

One possible option was designing the pretreatment method that could degrade limonene and produce sugars simultaneously. As envisioned, orange peel composition after oxidation did not show residual limonene. We confirmed that our method was successful by quantifying the limonene content in our samples before and after oxidation using Dart-MS analysis (Figure 5). Our results were in line with previous reports where limonene oxidation and its oxidized products characterization were carried out in the presence of ozone [45,46,47]. Key monomeric oxidation products reported by these studies were levulinic acid, 4-acetyl-1-methyl cyclohexene, limonene oxide, 3-isopropenyl-6-oxo-heptanal, and oligomeric products which can be formed via a condensation reaction. Based on our mass spectral data, the formation of high MW compounds (m/z > 300) such as dimeric oxidation and oligomeric products are more likely due to the appearance of a peak at 371.313 m/z (Figure 5) after a superoxide radical reaction. It is speculated that high MW compounds would form via a limonene alkyl peroxyl radical reaction [45]. Despite the widespread use of DART-MS, its capabilities to analyze a wide range of chemical masses is not well known. The only goal for the use of DART-MS in this work was to quickly evaluate the limonene degradation/oxidation. 

In a parallel study, the possible inhibitory growth effects posed by 0–5 *w*/*v*% of oxidized orange peel liquor added to LB culture medium on *E. coli* was investigated within 7 h of culture time (illustrated in Figure 6a). *E. coli* in the presence of oxidized orange peel liquor were able to maintain an acceptable growth rate (OD_550_ of 0.9 after 7 h incubation) and reached 1–3 g/L of cell dry mass with 1–13 *w*/*w*% PHB during fermentation. The cell growth in the presence of 3 *w*/*v*% oxidized orange peel liquor was comparable to the control curve. Good cell viability in the presence of up to 3 *w*/*v*% oxidized orange peel (4% increase in cell growth compared with the LB media) indicates that the obtained liquor contains carbon sources that can improve bacteria growth rate and that the inhibitory compounds (acetic acids, furfural, and HMF) were not at concentrations that negatively impacted growth kinetics, thus an additional step to remove inhibitory compounds was not necessary. However, increasing the oxidized orange peel concentration above 3 *w*/*v*% resulted in an inhibitory effect on the cell growth (Figure 6a). At a concentration of 5 *w*/*v*% of treated orange peel, a 20% decrease in cell viability was observed. The cell growth in the presence of 3 *w*/*v*% oxidized orange peel liquor is significant for the indicated time frames (5.5–7 h) compared with 5% OX-OP cell growth (Figure 6b). The lower concentrations of oxidized orange peel liquor (0.5% to 1.5%) showed statistically significant differences compared with the 5% OX-OP sample. There was no significant difference between 0% OX-OP and 5% OX-OP.

In addition, to study the suitability of acid hydrolyzed orange peel liquor as a carbon source and to compare the acid hydrolyzed orange peel with oxidized samples, a cell viability study was carried out (Figure 7). It was noted that a high concentration (3 *w*/*v*%) of the acid hydrolyzed orange peel caused a 50% drop in bacteria growth due to the existence of inhibitory compounds and limonene. 

Finally, to study different growth patterns of JM109 in the presence of 2 *w*/*w*% glucose, fructose, simulated orange peel sugars, and oxidized orange peel liquor during the actual fermentation, cell growth was monitored and the OD_550_ was collected by taking out an aliquot every 6 h (Appendix A). The slight decrease in cell density in the first 12 h of fermentation is due to the addition of IPTG (T7 RNA polymerase inducer) as well as the carbon sources [48]. The logarithmic phase of cells supplied with the oxidized orange peel was compared with glucose, fructose, and the simulated orange peel sugar solution. The oxidized orange peel liquor was rich in nutrients, and thus was suitable for cell growth. Cells fed with all carbon sources reached a stationary phase on the first day; however, a higher cell density was achieved with oxidized orange peel presumably because the orange peel media favored additional minerals and other degraded compounds [41,49]. Similarly, it was pointed out before that orange peels extracts led to the highest PHB yield compared with other waste materials from Bacillus wiedmannii AS-02 OK576278 and it is considered as an inexpensive nutrient source for PHB production [50]. Overall, all four carbon sources attained continuous cell proliferation after 3 days of fermentation. 

### 3.5. Cell Dry Weight and PHB Content

The bacterial growth study was performed to choose the proper carbon source concentration for PHB production. Based on the results, it was found that 0.5 to 3 *w*/*v*% oxidized orange peel liquor kept the bacteria rate of growth comparable with the control (LB). In addition, previous reports addressed the better growth rate trend of bacteria in medium amended with 2 *w*/*v*% pure carbon sources (e.g., glucose and fructose) [51,52]. Consequently, for this study, the cells were supplemented with 2 *w*/*v*% carbon sources and cultured for PHB production for 24, 48, 72, and 96 h to study the effect of carbon sources and fermentation time on cell dry weight (CDW) (g/L) and PHB content. 

Cell dry weight and PHB content when *E. coli* JM109 were fed with pure sugar sources (Figure 8) was 1.97 g/L and 29 *w*/*w*% CDW for glucose, and 2.15 g/L and 27 *w*/*w*% CDW for fructose after a 72 h incubation, respectively. The similarity in the CDW and PHB content from glucose and fructose is not surprising as they are transferred into the cells via a similar mechanism and JM109 metabolizes both of them to pyruvic acid with similar rates [53]. The control group (no added sugar) produced 1.02 g/L CDW, with a 48% increase to 1.52 g/L by extending the fermentation time from 24 h to 72 h. However, the control group generated only a small amount of PHB (13–22 mg, 1.0–1.5 *w*/*w*% CDW, 0.06–0.11% production yield). Simulated orange peel sugar achieved dry cell weights between 1.28–1.987 g/L and between 88.78–91.16 mg of PHB (4.6–6.9 *w*/*w*% CDW, 0.39–0.69% production yield) over the different incubation times. In the case of oxidized orange peel liquor, cell dry weight increased significantly from 1.71 g/L to 2.93 g/L when the time of incubation increased from 24 h to 72 h. The corresponding PHB content was 136.6 mg and 322.4 mg, correlated to 7.9 and 10.9 *w*/*w*% CDW, respectively. The yield of PHB produced from oxidized orange peel liquor ranged from 0.68 to 1.97% which was significantly higher than simulated orange peel sugar. 

We have proven that our oxidation pretreatment provides sugars that can be converted to PHB while eliminating growth inhibiting substances. Future work on this system will be undertaken to optimize the conditions to boost yield including substituting the type of fermenters, bacteria strains, and supplementing the oxidized orange peel liquor with some additives [18,54].

Assessing the cell dry weight and PHB content (*w*/*w*% CDW) obtained from pure sugar sources against oxidized orange peel liquor, the oxidized orange peel liquor increased the cell dry weight, however, the PHB production per cell dry weight decreased. The high cell dry weight necessarily does not improve PHB production, in some samples the CDW was high, but no PHB was formed. Oxidized orange peels contain different sugars and constituents that are conducive to the high growth rate of bacteria. Oxidized orange peel liquor may contain cellulose, hemicellulose, free sugars, lignin, fatty acids, and oxidized products that can be fermented to biopolymers if the conditions are right [55]. The produced yield of PHB mainly depends on the appropriate selection of bacteria strains. Some strains can consume different carbon sources at the same rate [56], however, JM109 bacteria may not spontaneously convert hexoses, pentoses, and fatty acids to PHA derivatives. 

Extending the incubation time from 24 h to 72 h, we observed that PHB was constantly made by cells, that is, cell dry mass increased from 1.20 g/L (24 h) to 1.97 g/L (72 h) in the presence of glucose, and from 1.07 g/L (24 h) to 2.15 g/L (72 h) in the presence of fructose. In addition to an increase in CDW, the PHB content increased from 8.3% (24 h) to 29.0% (72 h) and from 10.7% (24 h) to 26.7% (72 h) when glucose and fructose were used as feedstocks. Interestingly, the cell dry weight and PHB content decreased to 1.49 g/L and 28.1% for glucose culture and 1.44 g/L and 20.81% for fructose culture when the fermentation continued from 72 h to 96 h. However, the trend was different for the oxidized orange peel in which increasing the incubation time from 72 h to 96 h improved both cell growth by 1.6% and the PHB accumulation by 20%. A similar production rate of PHA has been obtained with *H. campisalis* MCM B-1027 bacteria when agro-wastes are used as a carbon source [57].

The overall ANOVA table (Appendix A) confirmed that carbon sources and the time of incubation are both significant factors in CDW and PHB production and that the interaction is statistically significant (*p*-value < 0.05). The change in CDW and PHB content over time is significantly different for the different carbon sources (or the difference between the carbon sources depends on the time point).

### 3.6. Extraction and Characterization of PHB

#### 3.6.1. NMR and FTIR

PHB from all runs was extracted, and the structure of the obtained PHB from different carbon sources was confirmed by ^1^H-NMR and ^13^C-NMR (shown in Appendix A). The ^1^H-NMR spectra showed a doublet at 1.26 ppm (methyl (–CH_3_) group), a doublet of quadruplet at 2.49 and 2.58 ppm (methylene (–CH_2_) group), a multiplet at 5.24 ppm (methane (–CH) group), and the CDCl_3_ solvent reference peak at 7.26 ppm. 

The ^13^C-NMR analysis also validated the structure of the obtained PHB polymer. Peaks at 20.16, 41.19, 68.01, and 169.54 ppm were assigned to methyl, methylene, methane, and carbonyl groups, respectively, which were similar to the peaks related to commercial PHB [58].

The PHB functional groups were identified by FTIR analysis. The FTIR spectra (Appendix A) exhibited a carbonyl band at 1724 cm^−1^ and a C-O stretch at 1280 cm^−1^ (for the ester bonds) which are the most characteristic peaks of PHB. In addition, -CH_3_, -CH_2_, and -CH group bands appeared at 1380 cm^−1^, 1458 cm^−1^, and 2922 cm^−1^.

#### 3.6.2. Ultrastructural Studies of JM109

TEM images confirmed that JM109 cells, when grown in LB media without the addition of any exogenous carbon sources, barely formed PHB granules confirming the importance of carbon source addition to generating PHB (Figure 9a). Whereas, after 48 h of cell cultivation in the presence of glucose, simulated orange peel sugars, and oxidized orange peel liquor, the TEM images showed white spheres resembling the formation of PHB granules (Figure 9b). TEM images supported the formation of a small number of spherical PHB granules (<10 *w*/*w*% of cell dry weight) (two to three per cell) with a diameter of 250 to 350 nm. The formed granules in JM109 when supplemented with glucose were smaller in size compared with those that had been formed when simulated orange peel sugar or oxidized orange peel liquor were utilized as carbon sources. The diameter of spherical PHB granules at the early stages of cultivation can vary from 30 to 150 nm. Longer incubation times can trigger further PHB accumulation which is observed as larger size PHB granules [59]. By extending the time of incubation from 48 h to 72 h, although some cells contained large PHB granules (diameter of 500 nm) (fed with oxidized orange peel), predominantly the granules form of PHB did not keep its shape (Figure 9c). Due to granule morphological changes, PHB appeared as clusters. Individual PHB granules likely coalesced by extensive fermentation which changed their spherical shape [60]. 

#### 3.6.3. Molecular Weight Analysis of PHB by GPC 

GPC data are presented in Table 1. *Mw* of PHB can range from 10 to 3000 kDa and is an important factor for the polymer degradation rate and its mechanical strength. The high *Mw* PHB (600 kDa) with low polydispersity is preferred for commodity applications [61]. The *Mw* range depends on the strains of bacteria, the type of carbon sources and its concentration, the time of incubation, and the extraction methods [62].

After 24 h of fermentation, JM109 cells produced PHBs with ultra-high molecular weights based upon the carbon source: for glucose, 1700 kDa; for fructose, 2100 kDa; for simulated orange peel sugar, 2000 kDa; and for oxidized orange peel liquor, 1100 kDa. After a 48 h fermentation, however, the *Mw* values dropped for glucose and fructose based samples. However, *Mw* increased for simulated orange peel sugar (2700 kDa) and oxidized orange peel liquor (1900 kDa) after a 48 h fermentation. The reduction trend of *Mn* and *Mw* was observed for PHB samples produced from simulated orange peel sugar (*Mw* = 2400 kDa and *Mn* = 1900 kDa), and oxidized orange peel liquor (*Mw* = 1700 kDa and *Mn* = 1200 kDa) after 72 h. Increasing the time of fermentation to 96 h further decreased *Mw* for both simulated orange peel and oxidized orange peel liquor derived PHB. We speculate that some compounds in the oxidized orange peel liquor could act as chain transfer agents and cause the secretion of PHB oligomers into the culture media [64]. The secreted PHB oligomers could then undergo degradation via hydrolysis, or cleavage by endogenous esterases, lipases, or other enzymes thus leading to *Mw* reduction [65,66,67,68]. The products of the degraded polymers could then be used as a new source of starting materials for the bacterial culture and lead to further PHB production. 

The polydispersity of the isolated PHB synthesized by JM 109 was in the range of 1.86–2.31, with higher values associated with larger molecular weight polymers. The results demonstrate the transition from a narrower range of *Mw* in PHB toward the higher range of *Mw* with a longer fermentation time. A longer fermentation time results in additional monomer formation during PHB synthesis, that is, as the PHB chain length increases, it becomes increasingly difficult for it to fit into the bacteria synthase active sites and this leads to chain cleavage and higher polydispersity [65]. 

The oxidized orange peel liquor produced PHB with the *Mw* of 1900 kDa and a *PDI* of 2.09, indicating that, as a feedstock, it is a reliable carbon source for producing PHB.

#### 3.6.4. Differential Scanning Calorimetry Analysis (DSC)

The crystalline/amorphous structures of PHB can be highly transformed by the type of carbon sources and the time of fermentation. For instance, PHB-derived rice bran showed similar structural properties as the commercial PHBs, while changing the feedstock to glucose only formed PHB with a relatively low thermal stability and a high melting temperature [69]. Thus, DSC analysis was performed on the PHBs that were produced to assess the impact of carbon sources and incubation time on the thermal properties of PHB. DSC thermograms of produced PHB using glucose, fructose, simulated orange peel sugar, and oxidized orange peel liquor at designated time frames are depicted in Appendix A. The thermal properties of extracted PHB obtained from the four sugar sources are summarized in Table 1.

Glass transition temperature (T_g_), an important feature of polymer chain mobility, is a temperature between onset and offset in the DSC curve [58]. The T_g_ data for the PHB produced using glucose, fructose, and oxidized orange peel liquor as a feedstock was relatively low, varying from −6.80 °C (24 h fermentation) to 6.8 °C (72 h fermentation) which is consistent with the literature reports where hexose-derived PHB was reported to shift the T_g_ to a lower temperature compared with pentose sugars [58]. The T_g_ reported in the literature for PHB varies greatly from −15 °C to 10 °C depending on the nature of the carbon sources and the experimental condition of PHB production. The ideal T_g_ for PHBs of industrial interest is 2.4–6.2 °C [70,71,72]. The highest T_g_ values were obtained when the carbon source was the simulated orange peel sugar with a T_g_ of 20.8 °C after 72 h which is outside the optimal T_g_ range. T_g_ values can be affected by *Mw* and *PDI*, a PHB polymer with a high *Mw* generally exhibits higher T_g_ values, while broad polydispersity of the polymer shifts the T_g_ peaks to lower temperatures [73]. 

The cold crystallization temperature (T_c_) was found to be in a range of 85.2 °C to 105.2 °C for all PHB samples produced from different sugar sources regardless of the type of carbon source and the length of fermentation. The T_c_ value correlates to the polymer chain configurations; if the chains pack closely, the temperature values shift to the higher temperature on the DSC curve. Further, the narrow crystallization peak indicates the crystals have a homogeneous size distribution (Appendix A). The T_c_ observed for PHB samples derived from glucose after 24 h and 48 h fermentation, simulated orange sugar peels after 72 h, and the oxidized orange peels after 24 h and 96 h fermentation had wide crystallization peak representing the formation of crystals with lower perfection and a larger size distribution [74].

The melting temperature (T_m_) for PHB varies from 167.4 °C (oxidized orange peel liquor, 24 h) to 174.8 °C (simulated orange peel sugar, 96 h), which is in the range of the values reported in the literature [75]. T_m_ is dependent on polymer *Mw* and, at a higher *Mw*, the higher melting point can be achieved. As observed in our GPC study, increasing the time of fermentation from 24 h to 48 h increased the *Mw* of polymer, irrespective of the feedstock, thus their T_m_ shifted to higher temperatures due to a decrease in chain mobility [31]. 

The degree of crystallinity determined for the produced PHBs ranged from 12.3% to 63.0% with the lowest for the PHB from oxidized orange peel liquor after a 24 h fermentation and the highest for fructose after 96 h fermentation. The PHB produced when glucose was used as a carbon source exhibited a degree of crystallinity between 27.3% (24 h) and 47% (72 h). Cells fed with oxidized orange peel liquor yielded PHB with a maximum 38.4% degree of crystallinity after 72 h fermentation. Low crystallinity is a sign that the generated polymer is both shorter and not a perfect polymeric chain. In general, the degree of crystallinity increased with incubation time: the longer the incubation, the higher the crystallinity. Depending on the PHB application, lower crystallinity can be advantageous; it imparts good mechanical properties and a better extensibility at break into the PHB which eases processability [76,77,78]. Lower crystallinity was also reported in the case of Poly(3-Hydroxybutyrate-co-3-Hydroxyvalerate) production [79].

All PHB produced in this study showed lower values of thermal parameters than the most reported formed pure PHB as noted in Table 1. Low thermal properties can be explained by impurities that may be trapped in the polymer, such as cellular-related hydrophobic materials (lipids, fatty acids), and that are simultaneously extracted by chloroform [76]. The degree of crystallinity is also affected by the presence of comonomers in the PHB polymer (copolymer of hydroxybutyrate and 3-hydroxyvalerate) [59,80,81]. According to a DART-MS study (data are not shown here), approximately 1% of the extracted PHB structure in this study contained 3-hydroxyvalerate.

## 4. Conclusions

Mild oxidation conditions were used to extract sugars from orange peels and degrade compounds with an inhibitory effect on bacterial growth. These extracted sugars can successfully be used as a nutrient source for biofermentation. Low concentrations of superoxide radical anion degraded orange waste and limonene quickly and under mild conditions. The resulting liquor improved the cell viability of E. coli strain JM109 by 90–100% and led to 136–393 mg of PHB accumulation during a 24 to 96 h fermentation. The entire oxidation pretreatment process occurred in one pot, and no extra purification/removal steps were required. Further, the produced PHB was characterized and found to have a high molecular weight and exhibited ∼30–40% crystallinity. The formed PHB from a fermenter containing the oxidized orange peel liquor as a feedstock generated the PHB with the same properties as the one obtained from the pure sugar sources. The obtained oxidized orange peel liquor triggered PHB accumulation with good physical and thermal properties due to the high quantity of carbohydrates and minerals required for PHB production.

## Figures and Tables

**Figure 1 polymers-15-00697-f001:**
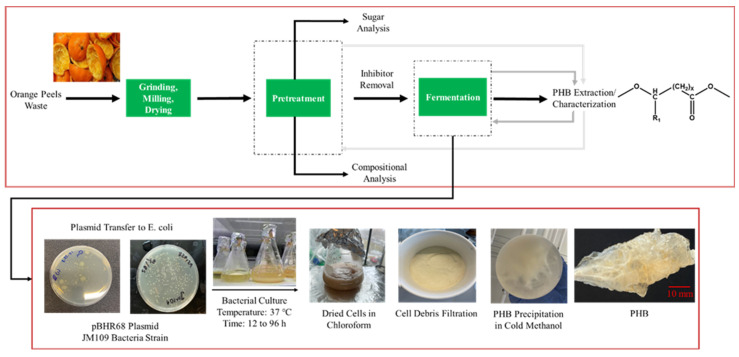
Schematic diagram for the experimental procedure of orange peel waste treatment, PHB production, and extraction.

**Figure 2 polymers-15-00697-f002:**
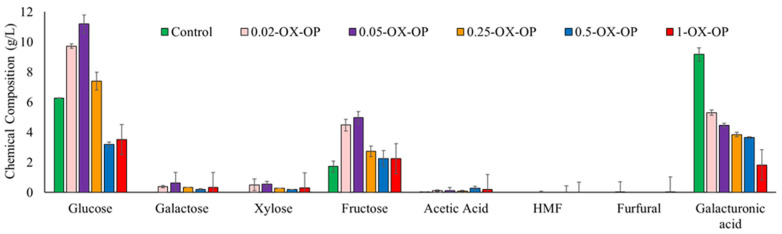
Sugar content in control and after 1 h of oxidation pretreatment by different concentration of oxidative agents.

**Figure 3 polymers-15-00697-f003:**
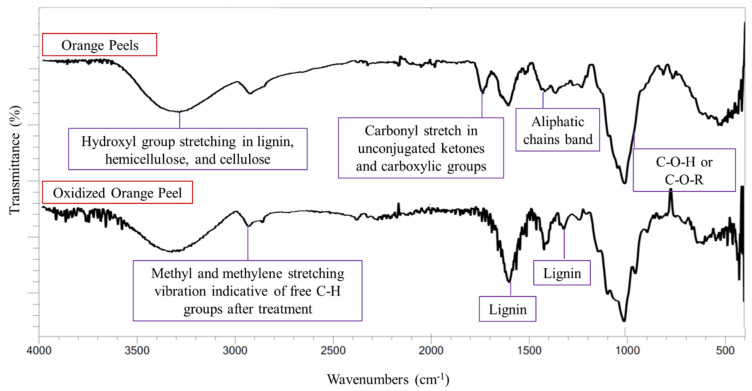
FTIR spectra of orange peels and oxidized orange peels.

**Figure 4 polymers-15-00697-f004:**
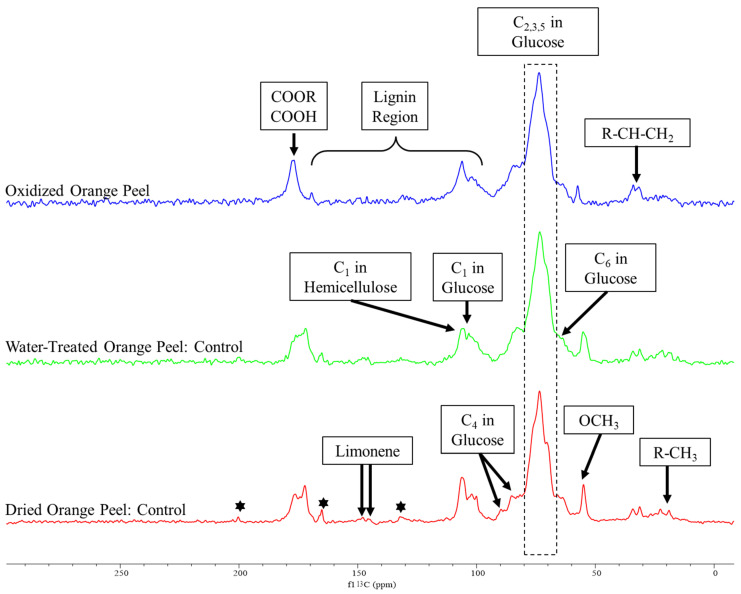
The solid-state ^13^C CP/MAS spectrum of dried orange peel, water-treated (to remove soluble sugars from dried powder), and oxidized orange peel. The stars indicate the presence of citral.

**Figure 5 polymers-15-00697-f005:**
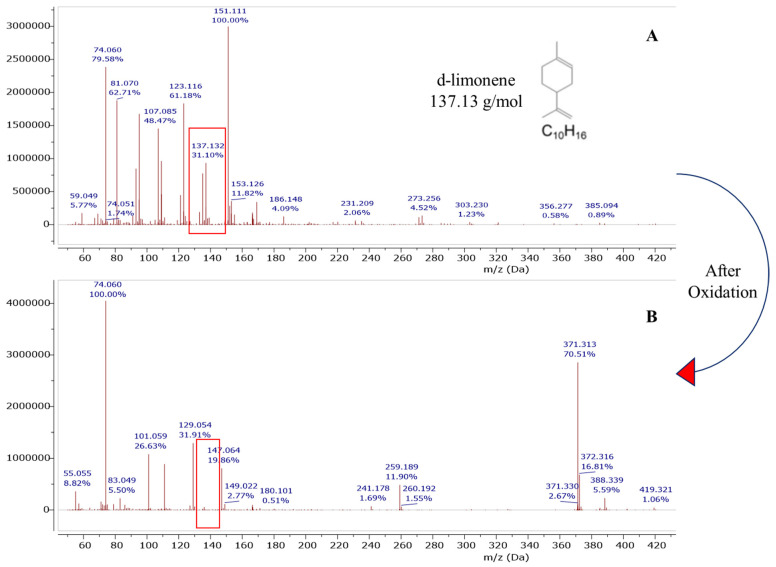
DART-MS study of limonene degradation before (**A**) and after oxidation (**B**) for orange peel liquor.

**Figure 6 polymers-15-00697-f006:**
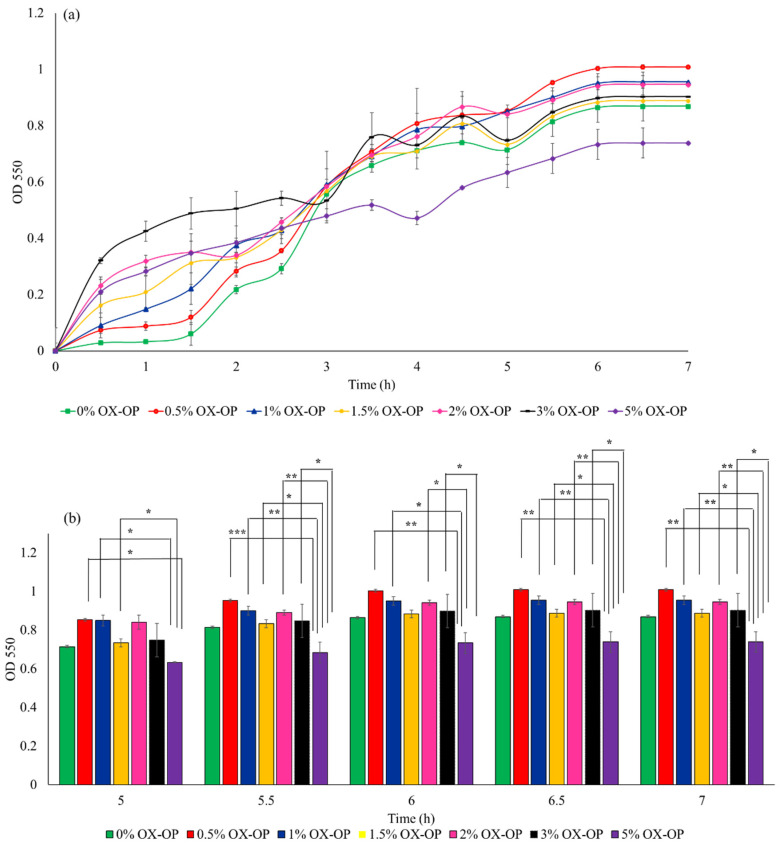
(**a**) Growth rate of JM109 E. coli in the presence of oxidized orange peel liquor (OX-OP). (**b**) Statistical analysis of growth rates by two-way ANOVA, (* *p* < 0.05, ** *p* < 0.01, *** *p* < 0.001).

**Figure 7 polymers-15-00697-f007:**
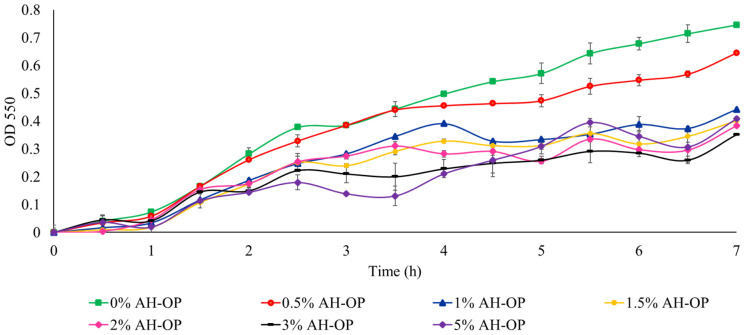
Growth rate of JM109 E. coli in the presence of acid hydrolyzed orange peel liquor (AH-OP).

**Figure 8 polymers-15-00697-f008:**
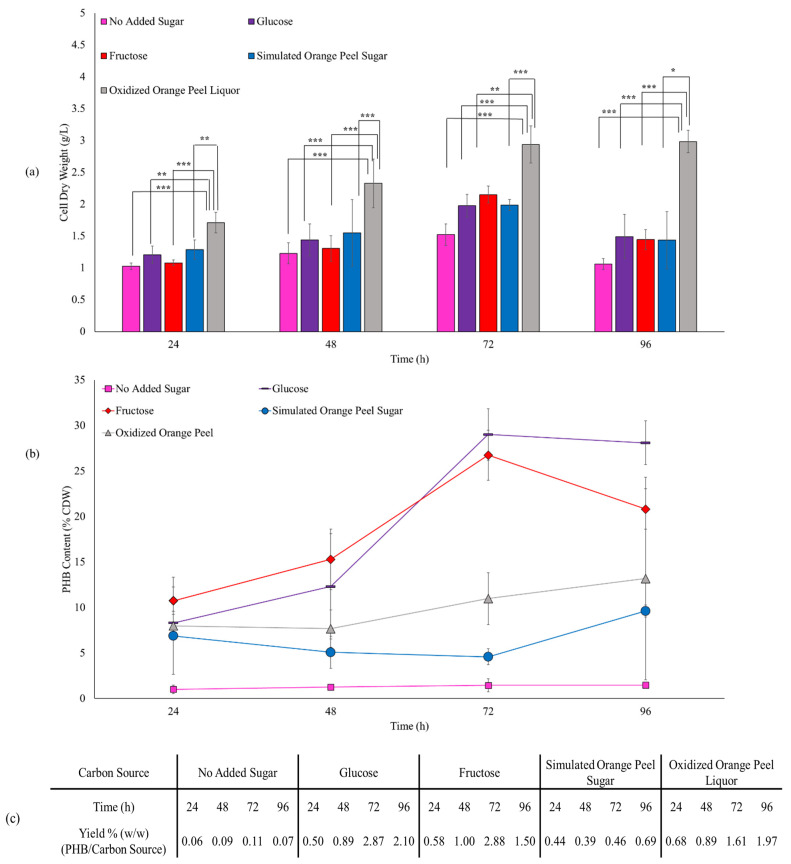
(**a**) Effect of various carbon sources on JM109 cell dry weight (bars) and (**b**) PHB content (lines). Error bars represent the standard deviation of uncertainty with 95% confidence interval. Statistical significance of data (* *p* < 0.05, ** *p* < 0.01, *** *p* < 0.001). (**c**) Production yields of PHB/substrate.

**Figure 9 polymers-15-00697-f009:**
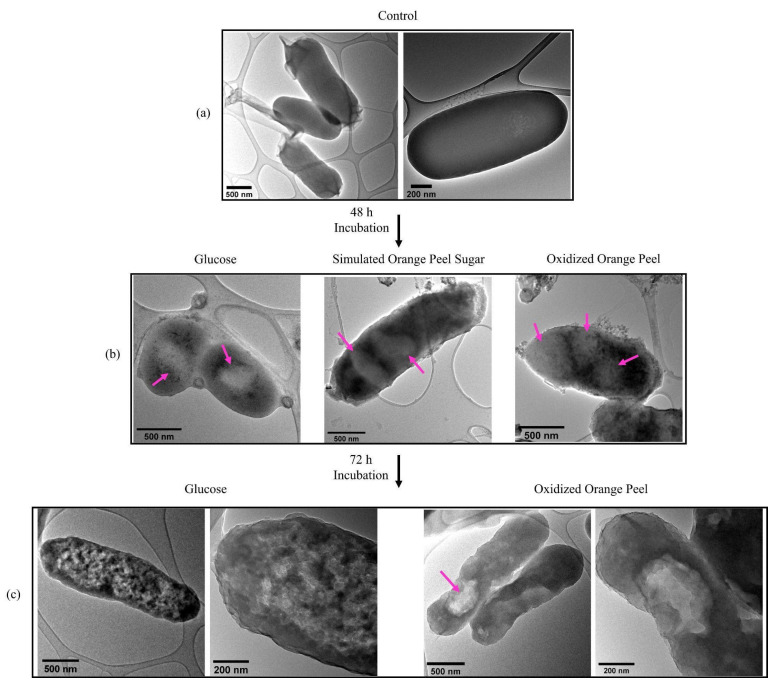
(**a**) Transmission electron micrographs of JM109 grown in LB without any carbon source (control) (depicted in two magnifications); (**b**) PHB granules in JM109 when fed with glucose, simulated orange peel sugars, and oxidized orange peel liquor after 48 h; (**c**) PHB clusters and granules in JM109 when fed with glucose (depicted in two magnifications) and oxidized orange peel liquor after 72 h incubation (depicted in two magnifications).

**Table 1 polymers-15-00697-t001:** Physical properties of the extracted PHB from different carbon sources obtained by GPC and DSC analysis. The degree of crystallinity was calculated by considering melting enthalpy of 100% crystalline PHB of 146 J/g [63].

Carbon Source	Time (h)	GPC				DSC				
		*Mn* (kDa)	*Mw* (kDa)	*MP* (kDa)	*PDI*	T_g_ (°C)	T_c_ (°C)	T_m_ (°C)	∆H_m_ (J/g)	X_c_ (%)
Glucose	24	890	1700	1400	1.91	−4.5	96.0	170.3	39.8	27.3
	48	580	1200	800	2.07	−0.8	98.5	172.3	40.9	28.0
	72	750	1600	1200	2.13	3.0	97.7	171.3	68.7	47.0
	96	1180	2500	2000	2.12	2.0	91.5	174.0	63.9	43.8
Fructose	24	1040	2100	1700	2.02	2.6	98.4	172.6	68.7	47.0
	48	570	1200	800	2.11	0.7	100.0	173.4	65.1	44.6
	72	1020	1900	1500	1.86	1.2	100.4	173.5	66.6	45.6
	96	990	2000	1500	2.02	−2.9	97.7	173.9	92.0	63.0
Simulated Orange Peel Sugar	24	960	2000	1600	2.08	8.6	105.0	172.3	25.0	17.1
	48	1170	2700	2400	2.31	17.4	105.2	174.3	31.4	21.5
	72	1170	2400	1900	2.05	20.8	85.2	169.1	24.2	16.6
	96	1010	2200	1700	2.18	18.0	102.5	174.8	50.8	34.8
Oxidized Orange Peel Liquor	24	540	1100	850	2.04	−6.8	92.0	167.4	17.9	12.3
	48	910	1900	1500	2.09	4.5	99.3	169.2	43.7	29.9
	72	880	1700	1200	1.93	6.8	97.9	170.0	56.1	38.4
	96	730	1500	1000	2.05	3.7	87.4	172.6	53.5	36.6

## Data Availability

Data is available upon request.

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
