# Peer review of "One-Step Oxidation of Orange Peel Waste to Carbon Feedstock for Bacterial Production of Polyhydroxybutyrate"

_polymers, 2023, doi:10.3390/polym15030697_

Round 1
Reviewer 1 Report
The manuscript describes pretreatment of orange peels to remove an inhibitory compound for the bacterial growth and then formation PHB/PHA using an engineered E. coli strain. Overall, the study is well designed. However, changes are required for better presentation an understanding of the manuscript.
1. Title: Bioplastic is a finished product. As authors did not attempt to finish their PHA/PHB so the word should not be included in the title.
2. It is important to use an evidence-back and uniform style of describing the product as PHA or PHB.
3. Do not repeat the words used in title as keywords
4. Correct in text citation. Refer the papers within a sentence
5. The introduction needs complete rephrasing; for instance:
a. -…biobased industry based on…
b. -…biological organisms… (all organisms are biological)
c. -While referring to biological oxidation by termites and fungi, the authors should
d. state the names of the enzymes or the metabolic pathway involved in oxidation of complex materials.
e. -…wild bacteria…
f. -While referring to suitability of orange peels as substrate for PHB/PHA production, authors should state some other sources and a comparison can be drawn to acknowledge the suitability of this substrate
g. -The background of pretreatment of orange peels is vaguely described and not conceivable
6. The materials and Methods section does not include sufficient details to reproduce the protocols, such as:
a. -The organization of this section needs improvement
b. -The source of orange peels should be given first. Also state the variety of the organs as it affects the composition of the peels. It is important as authors have not estimated the constituents.
c. -The strain number and source of Engineered E. coli is not mentioned in the first section.
d. -The pretreatment method needs to be elaborated.
e. -Some sections have been repeated for no reason
f. -Fig. 1 presents protocols that have not been given in the paper.
7. Results and discussion:
a. -What could be the possible reason for incomplete removal of limonene?
b. -Discuss how cellulose, hemicellulose was oxidized using K2O
c. -Apparently the growth rate in fructose containing medium was higher than the glucose containing medium. Discuss this observation
d. -Discuss variation in molecular mass of PHB in different sugars and its significance for the practical application of the bioplastic
Reviewer 2 Report
The article is very interesting and well written. Some minor issues must be addressed/clarified before publication acceptance. Please see the file attached.

Reviewer 3 Report
The main topic of presentation is devoted to the utilization of the fruits garbage as the eco-friendly procedure to improve municipal and agricultural environment. The authors have decisively demonstrated the multifaceted usefulness of orange peel treatment that enables the biotechnologists to fabricate more efficiently the biodegradable PHB as the principal representative of PHAs’ family. To settle the conformity of the biopolyester behavior to the analogous custom products, the set of physical techniques, such as NMR, TEM, NMR, DART MS, and etc., have been successfully used.
With the appropriate terminology and the reasonable argumentation, the manuscript idea promotes the trend of diminishing in the cost of PHB as constructive and special material to make its implementation more attractive. the biodegradable copolymers’ with improved mechanical, barrier, and biocompatible characteristics via the crystalline microphase pattern. The submission abstract reflects the general issues of the paper. The literature cited is quite relevant to this study and the illustrations (the table and the figures) are executed in unambiguous and accurate manner with the coherent interpretation.
Regularly, PHB is referred to the class of high crystalline polymers with the crystallinity degree within 50-80%. Please, specify what is the reason of moderate crystallites content (30-40%) comparing with regular PHB samples as films or fibers.
In Conclusion section, L18, the 4th and 5th lines: The sense of the sentence “Superoxide radical anion degraded orange waste and limonene at low concentration, short reaction time and mild conditions.” is intuitively understandable, but its grammatical structure should be essentially corrected.
Summarizing the Reviewer's opinion, it is worth recommending this manuscript for following Edition performance after making the above minor amendments.
Reviewer 4 Report
The article "Conversion of Orange Peel Waste to Polyhydroxybutyrate Bio-plastic" describes the valorisation of an industrial waste in bio-production of a plastic. It is a valuable study that can be published after authors address the following problems:
Some recent literature should be discussed in introduction: doi: 10.3390/Polym14101974; doi: 10.3390/Molecules26185594; doi: 10.3390/microorganisms9112395
In figure S6 the authors can assign the asymmetric and symmetric C-H vibrations from methyl and methylene on each peak, instead of middle one. A ratio of peak intensities from ~2980 and ~2930 (asymmetric C-H for CH3 and CH2) will give the ratio between those two groups in the PHB, which should be near the theoretical value.
For figures S7-S10 authors must provide a colour legend: what is blue line, what is red line, and what is green line (not everyone will understand that blue is the first heating step, green is the cooling step and red is the second heating step). Which is the value for melting enthalpy of 100 % crystalline PHB, and from which source it was taken?
Authors are encourage to read:
https://www.mt.com/de/en/home/supportive_content/matchar_apps/MatChar_UC232.html
and see if it is not more advisable to report the melting temperature as the onset one, instead of the peak value. Are sample considered impure and melting peaks broad?
Conclusion section must be reworked to reflect the heuristic of the study. Underline the novelty and advantages of this research, with actual numbers.
Reviewer 5 Report
The present work deals with the production of PHB from orange peel. In my opinion the topic is very appealing, and the paper is very well written, only some minors from my side:
- I suggest modifying section 2 for sake of clarity: in Paragraph 2.1, two sentences are repeated; Paragraphs 2.2 chemical formula, CAS and compounds concentration should be included for all the chemicals; Paragraphs from 2.3 to 2.5 can be merged in one as well as from 2.6 to 2.8. Please avoid in paragraph 2.9 all the subparagraphs.
- Figure 2, please avoid the legend explanation in the figure caption, but consider adding a specific table.
- Figure 5 should be more large
